# From the Initial Celebration to the Current Disappointment, the Evolution of the Internet beyond Determinisms

**Ezequiel Ramon-Pinat** [1,*] and **Ludovico Longhi** [2]

1 Comress-Incom Research Group, Autonomous University of Barcelona, Vinya st., Bellaterra-Cerdanyola del Vallès, 08193 Barcelona, Spain
2 Autonomous University of Barcelona, Vinya st., Bellaterra-Cerdanyola del Vallès, 08193 Barcelona, Spain
* Correspondence: ezequiel.ramon@uab.cat

**Abstract:** In the first phase, the eruption of the Internet was embraced by academics who saw it as a way of involving young people in politics who had suffered from disaffection and rejection. They emphasized its emancipatory, horizontal, and participatory qualities. Decades later, a wave of disenchantment, apathy, and rejection of platforms has swept through the academy. The previous generation of technological determinists, who welcomed it with open arms, left the arena to their counterparts: the ones that claimed that we had no chance of participation and perpetuated industrial age exploitation. In this article, we will present the two opposite visions, but first, we will briefly review the Internet's beginnings, its motivations, and its technical characteristics in order to better understand the two antagonistic positions.

**Keywords:** social media; political participation; pseudo-participation; technological determinism; critical theory 2.0

## 1. Introduction

In recent years, academia has taken a pessimistic view of social networks and their negative impact on society. Various negative effects such as the 'bubble filter', 'echo chambers', and 'toxic speech' have been identified. The fragility and influence of bots and manipulation during the US election that was won by Donald Trump were highlighted by the Cambridge Analytica scandal. Previously, there was a dominant opposing rhetoric that emphasized the emancipatory potential and the possibility for anyone to become a producer and reach a global audience.

Interpersonal communication platforms have been referred to using various terminologies from the beginning. Several terms have been cited, including social media, participatory media, social networks, digital social networks, Web 2.0, and participatory web. Initially, there was no consensus among academics due to its recent introduction as a subject of research. However, the lack of consensus arises mainly from the different approaches and paradigms employed to analyze it. In sociology, 'social networks' are defined as forms of support that individuals in groups can utilize to navigate adverse situations and meet basic needs. In the past, these networks took the form of personal contacts that provided emotional and practical assistance for activists.

By the late 1990s, communication researchers were beginning to take an interest in the Internet. Academics then turned their attention to the Internet, analyzing it in great detail. The authors concentrated on the legislation and regulations governing it, the technical makeup, and the utilization of this system. This focus was to the detriment of the application of conventional media effects theories (Kim and Weaver 2002). After two decades, a need still exists for a comprehensive and fundamental network theory that addresses its impacts and effective enhancement, coupled with the progress of fresh concepts and theories. Additionally, current communication theories are expected to

elucidate and project swift advancements (Fuchs 2011, 2018; Kim and Weaver 2002; Lovink 2004, 2016).

Fuchs et al. (2010) argue that it is crucial to develop not only a social theory of the Internet but also a critical social theory of the network. This theory can aid in comprehending how computing, including the Internet and the World Wide Web, can advance humanity's situation, promoting a better world. The authors dispute the claim that the Internet has become more social, asserting that it requires a critical assessment. They argue that this avenue is essential to enable academics and the public to understand sociality and social behavior on the Internet more accurately (Fuchs et al. 2010).

The lack of a dedicated sociology for Web 2.0 means that most definitions are derived from marketing or are still unresolved. On the other hand, the postulates are called for the development of a new theory that adapts to the new characteristics. They are claimed from a critical paradigm typical of the first Frankfurt School in the most innovative cases, and even from the most orthodox Marxism, rejecting phenomenology or empirical social research of the Web. Under this perspective, Fuchs, Hofkirchner, Schafranek, Raffl, Sandoval, and Bichler assert that knowledge is a threefold dynamic process encompassing cognition, communication, and cooperation (Fuchs et al. 2010).

## 2. The Transition from Web 1 to 2.0

The term 'Web 2.0', coined by Dale Dougherty and popularized by Tim O'Reilly, facilitated the emergence of a new era of the Internet. Users' activities shifted from replicating hyperlinked content, and in his document 'What Is Web 2.0', O'Reilly described the nascent changes it was undergoing (O'Reilly 2005). This shift to a new phase began in the late 1990s, although it was not until the start of the new millennium that it gained momentum.

O'Reilly's study, delving into computer science, focused mainly on the enhanced technical aspects of the web, including lightweight programming and the utilization of the Internet as a platform. Nonetheless, he also contemplated the social and cultural ramifications attendant to these qualities. Escalona (2013) and Olsson (2013) suggested that Web 2.0 provides opportunities to improve user experience and use collective intelligence.

This led to the adoption of the term "Web 2.0" to describe various online activities and applications. The use of mirror-like glazed icons also marked a new visual trend. However, the Internet at the time was not improved or polished, but rather focused less on the network itself and more on its usage, which gave rise to a generation of applications and businesses centered around the 'participatory web' in the form of blogs, wikis, and social networks, among others.

Tim O'Reilly foresaw key features of Web 2.0 applications, including the enablement of collective intelligence, the provision of interactive web-based services, and a shift from selling a product to providing an ever-evolving service (O'Reilly 2005). His predictions also touched on the growing significance of data and algorithms. These companies, which behaved distinctively from other firms operating on the static web or 1.0, experienced exponential growth. Flickr, Wikipedia, Digg, and BitTorrent constitute a limited selection of these (Madden and Fox 2006).

Although it is difficult to pinpoint the exact moment of the Internet's origin (Kleinrock 2010), the emergence of the early Internet was largely spurred by universities, large computer manufacturers, and the military. Key to laying the foundations for an open distribution architecture were hackers working within these institutions. In the second stage during the mid-to-late 1980s, a group of hackers from outside academia and small entrepreneurs came together, marking a golden age for cyberculture. This community was a blend of yuppies and hippies, characterized by an individualistic, libertarian, and anti-state attitude, as described by Lovink (2004).

The data network requirements were defined by engineers who established the management structure to enable its development and provided the necessary funding for ARPA implementation and deployment (Kleinrock 2010). The original four-node network and its extremely low speed, compared to today's connections, posed a significant challenge for

those pioneers. Cohen-Almagor argues that the notion of ARPANET (Advanced Research Projects Agency Net) as an attempt to construct a network that could withstand nuclear war is a widely held misconception (Cohen-Almagor 2013). Instead, we can place the Massachusetts Institute of Technology (MIT) and the University of California at Los Angeles (UCLA) as laboratories where the idea was first coined and later developed.

The second wave of developers held a similar reverence for open-source programs, including Unix, as their predecessors. Newly arrived members continued to contemplate the ethical implications of their direction and conducted extensive academic research. The third and subsequent phases saw an explosion in the size of the network and a tremendous increase in the number of connected users. The proliferation of these applications led to two increasingly prevalent patterns: "The growing popularity among users coincided with a greater centralization of online content in the hands of a small number of media conglomerates. This caused a (comparative) decrease in the proficiency of users" (Lovink 2004, p. 122).

The ties that were established in a globalized capitalist system were substantially reinforced. Where global energy networks, railways, and telegraphic and transatlantic networks made it feasible to reach far-off places, now greater distances can be covered in less time. Nevertheless, the organization was earlier mainly based on vertical production firms and hierarchical state apparatuses. Inter-nodal organization is not a solely contemporary occurrence, albeit networks' ability to introduce fresh actors and content into the social organization procedure has been enabled by the sudden emergence of communication technologies (Castells 2004, 2010).

## 3. Beyond Technological Determinism

When discussing social networks, the presence of ideas surrounding online communication, community building, and collaboration are frequently observed, either collectively or independently, despite any uncertainty surrounding definitions of phrases like free software and Web 2.0. The main issue pertains to identifying what is specifically novel and social, considering that the underlying technological frameworks of these networks and platforms existed before the conceptualization of these terms. Fuchs et al. (2010) comprehended the web as a techno-social system through their review of the literature on the dialectical construction of social theory and systems theory. Their aim is to bypass the perpetual debate between technological determinists and social determinists.

Manuel Castells introduced the concept of 'mass self-communication' (Castells 2009) to describe the new form of interactive communication enabled by the Internet, allowing the dissemination of messages from many to many. Some have labeled Castells a 'technological determinist'. However, he acknowledges the negative impacts on society resulting from this form of communication. However, the author concentrates on the shape of the node structure and the opportunities that the technology provides, as it can potentially reach a global audience and facilitate "self-communication" because the users themselves create the messages (Castells 2009). Technological advances have facilitated the formation of horizontal communication networks that have enabled continuous global connectivity through a variety of digital mediums, including blogs, podcasts, wikis, YouTube, Facebook, and X (formerly known as Twitter). These platforms have enabled the creation of virtual communities, establishing online social relationships (Moragas Spà 2011).

New electronic media are very powerful compared to the print media age, as they are extensions of our senses, including our central nervous system (McLuhan 1964). Digital media are unique in that they can be viewed as both a medium and a text. However, the text written in binary code (0/1) is not visible or decipherable to the end-user. Only the text displayed on the screen is visible and readable. Furthermore, if the differentiation between the medium and text is founded on the requirement that "in order for something to be a text it has to be signifying to a reader, then the text written with the letters 0/1 is part of the medium rather than of the text" (Brügger 2009, pp. 118–19).

It is important to recognize that the media is not solely a means of transmitting messages across long distances at a particular time. Furthermore, they alter the established reality by promoting particular social dynamics while suppressing or impeding others. Essentially, they shape the present state of society (Thompson 1995). Andrew Chadwick suggests the notion of "Political Technologies in Political Contexts" (Chadwick 2006, p. 19) as a means to overcome the Manichaean divide between technological and social determinisms. He disagrees with those who argue that technology is the sole determining factor and that understanding its effects can be accomplished by analyzing its inherent characteristics. He rejects the belief that a technology's properties are not linked to its political use. Instead, this approach proposes a more productive way of acknowledging the political properties of technology and situates its use within political contexts.

Technological autonomy, which refers to technical tools that operate without human intervention, has been a source of fear for generations. The Frankenstein metaphor, which describes technology that is out of control and acts against humans, has been revived in the form of AI and algorithms. At a less extreme level, technological autonomy can manifest in countless routines of daily life as undesirable aspects of a technological society erode human autonomy. We have become dependent on technology (Ellul 2004), lacking an independent sphere for activities that were once carried out without technological intervention.

Technological determinism overestimates the role of technology in society. It ignores or diminishes the fact that technology is embedded in it, and that it is people themselves who live in subjugation and rebel against power relations, triggering riots and revolutions, not technology. In a way, it ignores the political economy of events, while, at the same time, the emergence of new technologies tends to evoke a range of emotions that overlap with rationality. In the case of social media, it is an expression of the 'digital sublime', at the point where cyberspace has become the ultimate icon of the technological and electronic sublime, glorified for its characteristics of our time and demonized for the depth of evil it can conjure up (Mosco 2011).

An alternative to escape technological and social determinism is to conceptualize the relationship between technology and society as dialectic. Society conditions the invention, design, and engineering of technology, and technology shapes society in complex ways. Society is conditioned by technology, not determined by it, and vice versa. Therefore, social conditions, interests, and conflicts influence the technologies that emerge, but their effects are not predetermined. Modern technologies are complex, interactive, and unpredictable. Society adopts these complex forms, which means that their effects can be contradictory. As complex systems with multiple elements and interactions, they are unlikely to produce unidimensional effects. Technology is an enabling and constraining medium that results from the society in which it takes place (Fuchs 2012).

Kerkhof, Finkenauer, and Muusses argue that utopian or dystopian visions of its impact are too extreme and propose a view from syntopia: "People's physical and social situation and history influence their actions and what they do and learn online spills over into their real-world experiences. Relationships are shaped and developed in an ongoing process that takes place both online and offline" (Kerkhof et al. 2011, p. 149). The idea that technology alone can explain its success or failure in shaping human behavior is flawed. Various situations lead to different outcomes, a fact that is often overlooked by overarching theories that focus solely on its technology: "Different contexts produce different outcomes, something that is repeatedly obscured by overarching theories of the internet centred on its technology" (Curran et al. 2016, p. 25).

According to Geert Lovink, Web 2.0 is characterized by high usability, by facilitating social exchange and by allowing users to publish images, videos, and texts through free publishing and production platforms (Lovink 2016). In this new situation, it is the consumers themselves who recommend through search and sharing, and not the professionals as in traditional media. Back then, the influence of algorithms was not so dominant and there was the illusion of no filter between those who broadcast content and those who receive it.

## 4. The Old-New Power

Marxism at the beginning of capitalism denounced exploitation in absolute terms, without considering workers as differentiated beings with their own attitudes and wills. In the same way, with a structuralist and forceful vision, a new wave updates the postulates of the Frankfurt school. Christian Fuchs calls this current 'critical theory 2.0'. He maintains that social networks do not constitute a sphere of political participation since the vast majority of conversation that occurs on them is about leisure (Fuchs 2011). When we visit a website, "we don't expect to be hit with malware any more than we expect to be poisoned at a dinner party; we trust that the links we click won't lead to sites that turn our computers into mini-panopticons" (Morozov 2012, pp. 146–47). Shoshana Zuboff warns about the danger of being under constant surveillance by cell phones. It is important for mental health to have space for intimacy that is not monitored (Zuboff 2019).

Andrew Chadwick affirms the role of the Internet as an inherently political set of technologies, subject to decisions made in highly political contexts. Corporations and governments determine the types of technologies that operate on network software and hardware, which, while inevitably resisted and challenged, are often able to determine the architecture that can subsequently be used to regulate behavior. This modulation can be progressively more effective if it operates through preventive or automated means (Chadwick 2007).

Power relations can be understood either in a dynamic vision, in which they crystallize in a relationship between two actors, or in a structural vision, which emphasizes the form they take at the grassroots level in each society. The relationship between power and the media is intrinsic because whoever has power controls the discourse and the media, just as whoever controls the media has a privileged position in power relations. The media and their owners seek to influence, impose their own agenda on public opinion, and in the case of private media, profit.

The Frankfurt School, through authors such as Theodor Adorno, Walter Benjamin, Max Horkheimer, and Herbert Marcuse, denounced the oppressive role of the state and capitalism in a linear influence from the top down. Today, given the diversity and abundance of media and the dominance of market logic, the trend is to reduce state intervention to a minimum. The existing regulation ensures the viability of the market. Regulatory bodies act as traffic controllers, ensuring that circulation and trade can develop as freely as possible. Faced with this panorama, the new critical current 2.0 calls for the opposite of its predecessors: greater state intervention to regulate and limit the power of mega-corporations such as Alphabet and Meta, which are considered public goods. It also calls for the transparency of algorithms, the 'secret formula of Coca Cola', and the dissemination of the aspect and importance they give to variables.

As noted above, despite the initial technophilia and subsequent disappointment of Critical Studies 2.0, there are several scholars who assert its role in community building and political participation. They emphasize the use of the Internet for entertainment, such as online games, and its contribution to democracy by providing new ways of accessing social capital. However, the heterogeneity of members sharing a virtual space and social tolerance are not limited to online games and can easily be generalized to other types of communities (Kobayashi 2010). The potential of the Internet is conditioned by the nature of its use. There are young people who use digital media for civic purposes, such as reading the news, joining groups, and discussing political issues. Moreover, the cost of mobilization is dramatically lower on the Internet, which can activate the repertoire of participation (Boulianne and Theocharis 2018). However, a few pejorative terms such as clicktivism, click-activism, slacktivism, or 'flash activism' have proliferated to denote the lack of commitment to participation (Treré and Cargnelutti 2014).

Advances in data storage and processing have enabled the application of social signal processing. In recent years, there has been a surge in monitoring emotional states, particularly positive ones. A social signal is a "communicative or informative signal that, either directly or indirectly, conveys information about social actions, social interactions, social

emotions, social attitudes and social relationships" (Poggi and D'Errico 2010). Understanding human behavior through emotions is crucial. The value of images, particularly selfies on Instagram, has been explored by Walsh and Baker (2017) and should be investigated in relation to context, social emotions, and strategies for expression and sharing. In the field of social attitudes, important issues include expressions of agreement and disagreement, as well as self-presentation and its effects on persuasion (Poggi and D'Errico 2010). The interpretation is not based solely on an on–off affect basis. Instead of identifying emotion-specific facial configurations, the cumulative result of a series of individual facial actions presents a challenge (Mortillaro et al. 2011).

The Internet has led to the disintermediation of news preparation, production, and dissemination. In the era of big data, the focus has shifted from the 'why' to the 'what', which has radically transformed the way we explain the world. The datafication of the network has enabled new contact relationships, allowing us to move from a society based on events to one based on the information that shapes our reality (Mayer-Schönberger and Cukier 2013). Social networks have a business model that involves selling personalized user data. Therefore, their primary focus is on accumulating as much data as possible. The only factors that matter are those that can be measured and the relationships that can be established between them. The objectives of platforms such as Facebook, Instagram, or X are different from ours. We do not wake up every day with a commitment to increasing the time we spend there or making the experience more intense. It mainly goes in the opposite direction. Users should develop the set of terms that we want to use to name these phenomena. If we do not do this, they will do it instead (Williams 2018).

## 5. The Pseudo-Participation in 2.0 Age

The Internet generated a wave of enthusiasm for its potential to usher in a digital democratic era based largely on the desire to reproduce virtual public spheres and that, by lowering the cost of participation (since anyone with a computer could connect without needing to move), would be universal. It was claimed that 'democratic governance could be significantly improved by the open and equal deliberation between citizens, representatives and policy makers that new information and communication technologies make possible' (Loader and Mercea 2012, p. 1).

Although this hope never materialized, a new wave of technological optimism subsequently accompanied the emergence of social media platforms such as wikis, blogs, Facebook, Twitter, and YouTube. This second generation of the Internet, with more democratic and participatory aspirations, proposed the displacement of the public sphere model by a perspective centered on the citizen network, which had the opportunity to connect its autonomous private political sphere with a variety of political spaces at its disposal (Loader and Mercea 2012). Unlike the previous one, it focused on the role of the citizen-user as an engine of democratic innovation through the creation of self-configured networks of active citizens involved in identity and lifestyle politics, according to the agenda of the new social movements.

In a pragmatic way, political parties used it for the first time to raise funds for the electoral campaign, as in the case of the Democratic Party in the United States. This activity is crucial because, from a liberal point of view, unlike elsewhere, there are no regulations limiting donations. The parties' tours of the various states, which used to be conducted in person, could now be accomplished virtually. It should be noted that the Democrats already had a tradition of mobilizing their base, so the participation networks already had a substrate with a consolidated structure in which to take root (Andrejevic 2007). Unlike their republican rivals, they have traditionally been more open to new forms of participation and technologies.

More recently, the Internet has seen the proliferation of forums and conversations where people of different identities, workers, citizens, or consumers, have found their space to discuss, compete, collaborate, or simply share thoughts. Despite this proliferation, when it comes to politics, deep concerns about the character of contemporary liberal

democracy emerge spontaneously, as well as long-standing concerns in political philosophy about the possibility of sweeping away representative democracy and establishing a direct democracy on its site: "the potential of new technologies to realize the dream of increased civic engagement to address one of the central problems of contemporary liberal democratic politics: apathy" (Chadwick 2006, p. 25).

While some have celebrated the emergence of forums as free public spheres of political deliberation, as a civic common in cyberspace, they have also been criticized for the low quality of interaction they generate and, more recently, for their tendency to reproduce a plurality of deeply segmented political associations. In this sense, Eli Pariser has coined the term 'filter bubble' (Pariser 2017) and Cass Sunstein the term 'echo chamber' (Sunstein 2001, 2017). According to Kakisina, Indhiarti, and Al Fajri, ideological polarization, discrediting opponents, emotional arguments, emphasizing the speaker's power, moral superiority, and credibility, and presenting seemingly irrefutable evidence to support the speaker's beliefs and reasons are all tactics used in polarized argumentation (Kakisina et al. 2022).

Instead of serving as spaces for deliberation and the exchange of views, associations with peers who share a similar vision tend to reinforce our points of view rather than challenge us with alternative perspectives. During political debates, it is common to discredit opponents through criticism, insults, and subtle strategies. This often involves focusing on negative aspects of the rival, such as their perceived lack of benevolence, competence, or leadership skills (D'Errico et al. 2012). Far from a rational ideas exchange, we do not know what others think, and polarization has soared. Citizens lose the ability to understand and empathize with those who do not think like them. Shared information spaces are conspicuous by their absence, so the democracy of understanding is being corroded.

Uncontrolled Internet use has led to the development of new pathologies, particularly among teenagers. Problematic Internet Use (PIU) is associated with psychosocial impairments, neglect of offline social relationships, aggression, poor self-control, and narcissistic traits. Loneliness and shyness are important aspects of youths' social adjustment. The need for belonging, as a fundamental social need, plays a role in social relationships and the development of aggression. In males, the abuse dimension of neglect and disinhibition is prevalent (Piko et al. 2017). Anonymity and volatile identities facilitate online flaming. Aggression is an outcome of several coinciding aspects, such as motivation, inhibition, and opportunities provided by specific online contexts. This holds true for the material in question, "where permissive anonymity is backed up by a motivation to aggress provided by the journalists and other commentators" (Laineste 2013, p. 41).

Nor can Web 2.0 be considered a fully participatory democracy since it does not extend beyond the political sphere to culture and the economy. Above all, it maximizes the development powers of an economic class that owns the web platforms and the power of extraction. It does not maximize the development potential of human beings; on the contrary, it dispossesses and exploits users and workers in order to accumulate capital. Christian Fuchs speaks of pseudo-participation and exploitation because if "knowledge is a social and historical product, new knowledge emerges from the historical heritage of knowledge in society and is in many cases produced cooperatively" (Fuchs 2011, p. 284). On the other hand, the desire to differentiate practices that are only nominally participatory and that can be exposed as forms of pseudo-participation is common to all schools of thought that have analyzed the issue (Carpentier 2007).

Beyond the significant process that underlies the articulation of participation, it transcends its limits and must be framed within a political-ideological debate. From this perspective, its definition is located in one of the many social fields that oscillate between minimalist and maximalist variations of democracy and politics. It is a struggle between two archetypal political-ideological models. At least in the initial phase, the strategy for dealing with this significant diversity is not related to the task of distinguishing between authentic participation and pseudo-participation (Carpentier 2007).

## 6. The Meaning of Interactivity in Social Media

Interaction is one of the characteristics associated with the quality of political participation, analyzed by both technophilic and technophobic currents, albeit with apparently different conclusions. Interactivity has been studied for a long time in the sociology of communication. It is seen as the relationship between two or more actors who adapt their behavior and actions to each other in each situation. More recently, research in this field has taken a turn, focusing on the interaction of people with new information technologies (ICT), largely influenced by computing and mainly aimed at improving the effectiveness of the interface of hardware and software with users. The emergence of the GPT chat has caused great concern among workers who fear losing their jobs. For this reason, it has provoked the reaction of the Writers and Actors Guilds of Hollywood. However, it has become clear that while it may be very important for some tasks, it cannot replace a human for others. Despite enormous dedication to the subject, the interaction between humans and microprocessors is unstable, error-prone, and largely undefined.

In the mid-1980s, it began to attract the attention of communication scholars, who began to investigate the nature of interactivity in computer-mediated communication. Rafaeli was one of the first researchers in this field to understand interactivity as an expression in which, in a series of communicative exchanges, the third or subsequent messages are related to the previous ones. He paid attention to the sequence in which messages are related to each other, and in particular, to the extent to which later messages relate to earlier ones, with an emphasis on computer-mediated groups (Downes and McMillan 2000).

In any case, interactivity is not a monolithic concept. In contrast, it is polysemous and dynamic. If the analysis is limited to how individuals perceive it in the context of computer-mediated communication, the additional concepts of role-taking and feedback emerge. That is, for a medium to be fully interactive, the roles of sender and receiver must be interchangeable, and, in addition, they must have control over their mutual discourse. There are authors who add the temporal component, ensuring that the modification of the form and content of the mediated environment occurs in real time, although the asynchronous characteristics of tools such as email, newsgroups, or social networks challenge this limitation of immediacy (Downes and McMillan 2000).

However, as with the debate between technological and social determinism, it remains to be seen to what extent the qualities of individuals who use media, such as passivity and interactivity, overlap since they are not qualities of the media themselves but of their users. However, it cannot be ruled out that the nature of some technologies, such as hypertext and its non-linearity, favors interaction beyond the individual character of those who use it or the popular idiosyncrasies of the society in which it is used, without falling into social determinism.

However, the concept of interactivity can be narrowed down even further to a pre-programmed response within a system, where the message we receive refers to the immediately preceding one or to a series of previously exchanged messages. Interactivity therefore exists in subject–subject communication, but also, as noted above, in the exchange between a subject and a technological device, where "in this second example, interactivity develops in the interface, which could be defined as the site of interaction" (Scolari 2008, p. 94).

Under the paradigm of critical studies of social networks, Mark Andrejevic proposes a repressive hypothesis for interactivity, under Foucault's slogan that where there is power, there is resistance:

> "Where there is resistance there are always new and realigned strategies for control. We might go so far as to propose an interactive repressive hypothesis: whenever we are told that interactivity is a way to express ourselves, to rebel against control, to subvert power, we need to be wary of power's ruse: the incitation to provide information about ourselves, to participate in our self-classification, to complete the cybernetic loop". (Andrejevic 2009, p. 41)

From a commercial perspective, two sides can be seen in the representation of interactivity: one, pointing out the way in which the top-down media model is being challenged, and another, embracing the greater possibilities of information management and manipulation. The former perceives the emerging power of an almost tyrannical and demanding interactive consumer, while the latter presents it as an opportunity to reinforce greater control, with a hyper-focus on advertising and monitoring based on the rationalization of the marketing process (Andrejevic 2009). The transition from 'Web 2.0' to the broader concept of 'Services 2.0' means a return to the initial characteristics of the tools and the opportunity for the content to adopt a layer of social functionality (Escalona 2013).

Through interactivity, it is possible to learn more about consumers in order to determine the best way to influence them. The video game industry, in particular, uses personalized advertising based on detailed monitoring of the game combined with demographic information. Not only do they bombard you with relevant ads but they also use the interactive and immersive nature of the game as a means of discouraging critical reflection. The flip side of this approach is that active engagement, rather than passive contemplation, encourages critical engagement. However, frenetic interactivity helps to mask forms of control, as the very invitation to interact is a technique for managing audiences and channeling their activities.

From a marketing perspective, 'interactivity is embraced not for the ways in which it encourages challenge to dominant messages and critical skepticism, but for the ways in which it fosters them' (Andrejevic 2009, p. 42). This notion that hyper-interactivity can frustrate critical reflection rather than the active participation that was intended can be applied to the field of communication-information, where the enormous mass of information circulating in the media contributes to this end. In this scenario, interactive media short-circuit reflection by challenging the authority of unidirectional, top-down media technologies. It means the opposition between critical interactivity and passive consumption.

## 7. Conclusions

When new media emerge, a mixture of distrust and illusion takes place. Through a contemporary lens, it is viewed by the conjunctural aims and necessity. In the case of the Internet and social media, in the first phase, its eruption was embraced by academics who saw it as a way of involving young people in politics, who had suffered from disaffection and rejection. They emphasized its emancipatory, horizontal, and participatory qualities. They highlighted the chance for everyone to be a producer and play an active role in opposition to television.

Decades later, a wave of disenchantment, apathy, and rejection prevailed in intellectual analysis. The previous generation of technological determinists, who welcomed it with open arms, has been displaced by the ones that claimed that we had no chance of participation and perpetuated industrial age exploitation. Both underestimate the active character of society, re-appropriation, and mechanisms of rejection. The media and technology condition users' behavior and facilitate it, but do not determine it. There is room for individual attitudes behind them.

The transition from Web 1 to 2.0 involves more than just increasing the speed and volume of data. It also brings about changes in individual behavior and social relationships, which can be facilitated or hindered by technology. While the ARPANET project, hacker logic, and the university environment may have initially influenced the character of the Internet, a logic of free market and exploitation has since taken over the entire environment. In an unequal power relationship, large platforms take advantage of universality to pay taxes in places where they can pay less, store data and servers where they are not regulated, and extract data where there is no opposition. This results in users being unable to access the benefits of relocation.

Rather than engaging in sterile battles about who determines who, whether technology determines society or vice versa, we can understand it as a constant negotiation between

several parties. Throughout recent history, we have observed technologies that were announced as definitive, such as web 3 or semantics, have had no impact on society. Conversely, others that were thought to be ending, such as books, radio, and cinema, have remained. However, the impact of technology is not solely determined by citizens. Social networks have taught us that we can only interact with what the algorithm presents to us.

The field of Internet studies has undergone a shift in focus from emphasizing user capabilities to highlighting the extraction of data by platforms from user activity.

## 8. Future Directions

Currently, the academic community is primarily focused on the impact of bots on elections, the geopolitical implications of data extraction and storage in relation to TikTok and China, and the expansion of surveillance to include facial recognition technology. This necessitates the establishment of ethical boundaries, determining what is permissible, what constitutes a violation, and what should be prohibited. The emergence of new forms of communication requires the redefinition of established concepts and the introduction of new concepts and tools, such as pseudo-participation and interactivity. By defining them, we are framing them as something that requires attention or as something humorous that poses no danger.

Future research should address the concentration of platform property and its significant power over states. Governments face great difficulties in limiting them due to their delocalization. Another challenge to be analyzed is the rise of Artificial Intelligence and its evolution. The Semantic Web or Web 3.0, which was claimed to be the next step a decade ago, is already established. However, there are still many unconnected devices. The scholar should also pay attention to this development.

Current research must continue to ask, just as the classical critical movement did from the Frankfurt school to the present day, how we can build a better society. However, there does not exist only one solution. A combination of proposals may be necessary, including interstate regulation rules, the deconcentration of property, and media literacy campaigns. Academic research should be directed towards all of these solutions. Not only should we focus on the most restrictive aspects, such as analyzing technological and social network initiatives, but we should also promote good practices and socially relevant recommendations.

**Author Contributions:** Conceptualization, E.R.-P.; writing—original draft preparation, E.R.-P.; writing—review and editing, L.L. All authors have read and agreed to the published version of the manuscript.

**Funding:** This research received no external funding.

**Institutional Review Board Statement:** Not applicable.

**Informed Consent Statement:** Not applicable.

**Data Availability Statement:** The datasets used during the current study are available from the corresponding author upon reasonable request.

**Acknowledgments:** The authors would like to express their gratitude for the valuable advice provided by the members of the Network for studies of culture, inequality and democracy, Department of Information Science and Media Studies, University of Bergen, Norway. It is worth mentioning that this article was written during a stay at that center.

**Conflicts of Interest:** The authors declare no conflicts of interest.

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
