# Peer review of "From the Initial Celebration to the Current Disappointment, the Evolution of the Internet beyond Determinisms"

_socsci, doi:10.3390/socsci13020099_

Round 1

Reviewer 1 Report

Comments and Suggestions for Authors

Though concerns about Platformization and massive use of AI technologies by the public are mentioned as an ongoing debate, some reflections about how some old concepts and new ones can contribute should be incorporated in the final epigraph. Just a couple of references about this ongoing debate are needed.

Author Response

- Reflections on the incorporation of old and new concepts in an extended final epigraph. Additionally, a 'Future Directions' section has been added after the Conclusions. 
- References regarding the debate have been added. 
(Changes are highlighted in red in the new version uploaded). 

Reviewer 2 Report

Comments and Suggestions for Authors

The presented review focuses on the evolution of the internet, which passed from, according to the authors, enthusiastic to a critical stance. The paper shows these two opposite visions, but first, it briefly reviews the Internet's motivations, and its technical features.

The paper is well written and manages to bring out the parable of studies on the effects and possibilities of the internet, highlighting new concepts, such as pseudo-participation and interactivity, the presence of bots, the geopolitical interest in data storage, and surveillance expansion, read from a critical and aware perspective.

To the references made by the authors, however, I would add the entire area of surveillance, including emotional states monitoring emerging from the literature on social signal processing (see, for instance, Poggi, et al (2010, October). Cognitive modelling of human social signals. In Proceedings of the 2nd international workshop on Social signal processing (pp. 21-26). Mortillaro, M., Mehu, M., & Scherer, K. R. (2011). Subtly different positive emotions can be distinguished by their facial expressions. Social Psychological and Personality Science, 2(3), 262-271.) emerge how they are identifiable and how the performance of new technologies is easily applicable. Furthermore, all the aspects of great interest relating to aggressive and discrediting behaviours remain indefinitely and can bend the participatory nature of online contexts with that more relating to the transgression of the ethical sphere (see for instance: Laineste, L. ( 2013). Funny or aggressive? Failed humor in internet comments. Folklore: Electronic Journal of Folklore, (53), 29-46., Piko, B. F., Prievara, D. K., & Mellor, D. (2017). Aggressive and stressed? Youth's aggressive behaviors in light of their internet use, sensation seeking, stress and social feelings. Children and Youth Services Review, 77, 55-61.). This sphere also has strong implications on classic media communication which is increasingly similar to the processes of polarization and denigration triggered by new media. (Kakisina, P. A., Indhiarti, T. R., & Al Fajri, M. S. (2022). Discursive strategies of Manipulation in COVID-19 political discourse: the case of Donald Trump and Jair Bolsonaro. SAGE Open, 12(1), 21582440221079884.; D 'Errico, F. et al. (2012). Discrediting signals. A model of social evaluation to study discrediting moves in political debates. Journal on Multimodal User Interfaces, 6, 163-178.).

With the addition of these other aspects, I think this review can be considered for publication.

Comments on the Quality of English Language

good quality of the used language. 

Author Response

  • References of surveillance, including emotional states monitoring emerging from the literature on social signal processing, were added.
  • Aggressive and discrediting behaviours in the participatory nature of online contexts was treated with the suggested texts.
  • Processes of polarization and denigration triggered by new media were added. 
    (Changes are highlighted in red in the new version uploaded). 

Reviewer 3 Report

Comments and Suggestions for Authors

The article is interesting. Its content corresponds to the topic. However, the article needs to be extended. The analysis contained in Sections 2,3,4 and 5 is insufficient. The indication of future research directions is insufficient: "Future research directions should address the high concentration of platform property, and the extreme power they have against states." The authors should expand this area. The authors need to expand the article to include other articles. The research they have done is based on too few publications. 

Author Response

  • The articles was extended from 5355 to 7253 words.
  • The analysis contained in sections 2,3,4 and 5 was expanded.
  • A 'Future Directions' section has been added after the Conclusions.
  • Conclusions were expanded.
  • References were added. 
    (Changes are highlighted in red in the new version uploaded). 

Round 2

Reviewer 2 Report

Comments and Suggestions for Authors

The paper has been improved and it can be accepted at this stage.